# Vocabulary-Guided Gait Recognition

**Panjian Huang[1], Saihui Hou[1]***, **Chunshui Cao[2], Xu Liu[2], Yongzhen Huang[1,2]**
[1] School of Artificial Intelligence, Beijing Normal University
[2] WATRIX.AI

*"The way we extract gait features depends a lot on how we understand a gait."*

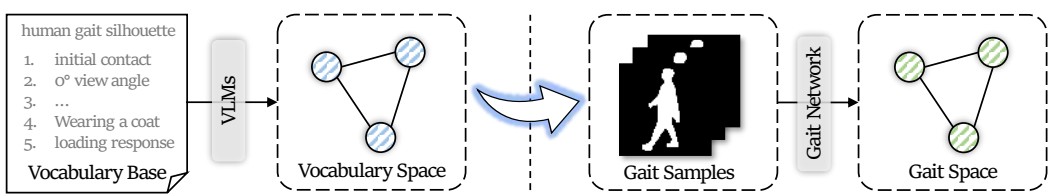

Figure 1: Vocabulary-guided gait recognition aims to explore gait concepts through human vocabularies with VLMs where the vocabulary features enable to guide the gait network learning. Specifically, the universal vocabulary space (*e.g.*, initial contact) can guide the gait network to derive the corresponding semantic gait features, thereby yielding a more universal gait space for practicality.

## Abstract

**What is a gait?** Appearance-based gait networks consider a gait as the human shape and motion information from images. Model-based gait networks treat a gait as the human inherent structure from points. However, the considerations remain vague for humans to comprehend truly. In this work, we introduce a novel paradigm Vocabulary-Guided Gait Recognition, dubbed **Gait-World**, which attempts to explore gait concepts through human vocabularies with Vision-Language Models (VLMs). Although VLMs have achieved the remarkable progress in various vision tasks, the cognitive capability regarding gait modalities remains limited. The success element in Gait-World is the proper vocabulary prompt where this paradigm carefully selects gait cycle actions as Vocabulary Base, bridging the gait and vocabulary feature spaces and further promoting human understanding for the gait. **How to extract gait features?** Although previous gait networks have made significant progress, learning solely from gait modalities on limited gait databases makes it difficult to learn universal gait features for practicality. Therefore, we propose the first Gait-World model, dubbed $\alpha$-**Gait**, which guides the gait network learning with vocabulary knowledge from VLMs. However, due to the heterogeneity of the modalities, directly integrating vocabulary and gait features is highly challenging as they reside in different embedding spaces. To address the issues, $\alpha$-Gait designs Vocabulary Relation Mapper and Gait Fine-grained Detector to map and establish vocabulary relations in the gait space for detecting corresponding gait features. Extensive experiments on CASIA-B, CCPG, SUSTech1K, Gait3D and GREW reveal the potential value and research directions of vocabulary information from VLMs in the gait field.

## 1 Introduction

Gait recognition aims to identify individuals based on walking patterns across complex covariates, *e.g.*, cross-view and cross-clothing scenarios [1]. As fundamental paradigms, appearance-based

---

*Corresponding Author

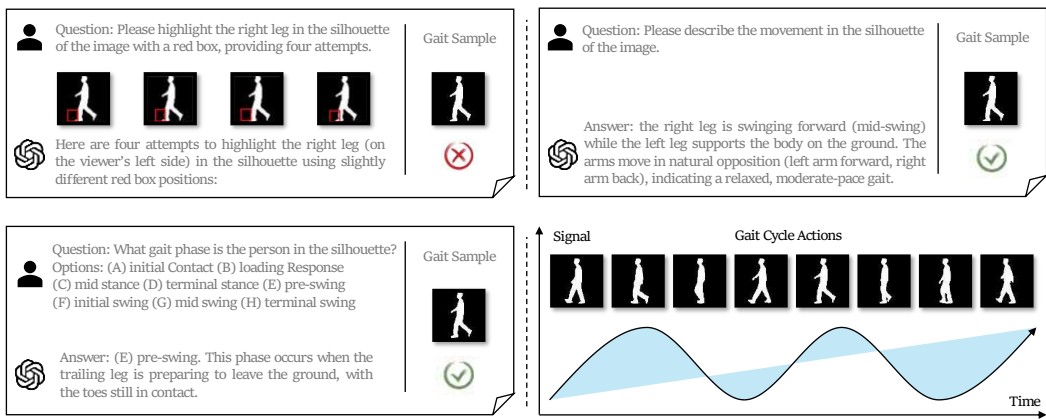

Figure 2: **Up**: In the silhouette modality, VLMs struggle to identify fine-grained details (*e.g.*, localizing the right leg), yet remain sensitive to basic walking patterns, which rely on overall structure. **Down**: The success element of Gait-World lies in leveraging gait cycle actions as the vocabularies to bridge the gap between VLMs and gait modality.

gait networks [2, 3, 4, 5, 6, 7, 8, 9, 10, 11], which treat a gait as human shape and motion information, typically take images as input (*e.g.*, silhouettes, parsing and optical flow) and use CNNs or Transformers to capture local and global spatio-temporal information. Model-based gait networks [12, 13, 14, 15, 16, 17, 18] treat a gait as the human inherent structure. These methods take points as input (*e.g.*, keypoints, meshes, and heatmaps), and typically apply GCNs or Transformers to extract local and global relations among points and edges.

Despite the significant progress made by these paradigms due to their efficiency, the gait research still faces two main problems: **(i) Ambiguous gait concepts** (*e.g.*, shape, motion and structure) cause that researchers either possess advanced knowledge yet cannot fully apply it due to underdeveloped gait networks, or have feasible insights but cannot verify whether the network truly works as intended. **(ii) Constrained gait networks** rely solely on the gait modality and the limited gait databases, struggling to learn universal features for the real-world scenarios. For example, appearance-based methods suffer from drastic appearance variations under cross-clothing conditions, model-based methods heavily depend on the accuracy of upstream pose estimators, and gait databases often encounter sparse-view [19] and cloth-imbalance [20] problems. Therefore, we naturally ask: ***What is a gait and how to extract gait features?*** Considering that human vocabulary inherently possesses interpretability and semantic guidance, we introduce a new gait paradigm and network to answer:

**Vocabulary-guided gait recognition.** The primary goal is to harness human-defined vocabulary with Vision-Language Models (VLMs) to explore gait concepts, thereby promoting gait networks and providing researchers more informative feedback. Inspired by *"The limits of my language mean the limits of my world."* from Ludwig Wittgenstein, we name the paradigm **Gait-World** shown in Figure 1. Gait-World consists of Vocabulary Base, VLMs, and Gait Network. Researchers provide basic knowledge as the Vocabulary Base, from which VLMs extract the vocabulary features to serve as priors that guide the gait network in learning corresponding gait features. However, integrating human vocabulary knowledge into gait modalities via VLMs is non-trivial because publicly available training data for VLMs rarely include gait samples. As shown in Figure 2, VLMs (*i.e.*, GPT-4o[1]) often struggle to identify fine-grained details (*e.g.*, hands or legs) in silhouettes. Nevertheless, we observe a phenomenon where VLMs remain sensitive to basic walking patterns. We adopt the clinically defined eight-phase gait cycle as a minimal, complete set [21, 22], and VLMs accurately recognize gait cycles from silhouettes, aligning with *"Gait serves as a walking descriptor."* Therefore, the success element in Gait-World is the proper vocabulary prompt where this paradigm carefully selects gait cycle actions as Vocabulary Base, bridging the gait and VLM spaces and further promoting human understanding for the gaits.

**$\alpha$-Gait.** Towards vocabulary-guided gait recognition, we introduce the first Gait-World model, $\alpha$-Gait, which leverages vocabulary knowledge from VLMs to guide gait representation learning. Specifically,

[1]https://openai.com/

due to the modality heterogeneity, directly integrating vocabulary and gait features poses a challenge, as they reside in distinct embedding spaces. To address this, $\alpha$-Gait firstly employs the Vocabulary Relation Mapper that maps the vocabulary feature into the gait space and establishes vocabulary relations. Then, the Gait Fine-grained Detector queries the gait features with the vocabulary guidance, extracting corresponding semantic gait features for recognition.

Our main contributions can be summarized as follows:

• We introduce a novel paradigm Vocabulary-Guided Gait Recognition, dubbed Gait-World, which applies human vocabularies with Vision-Language Models to effectively explore gait concepts, revealing the vocabulary value for the gait field.

• We propose $\alpha$-Gait in pursuit of the Gait-World, which designs Vocabulary Relation Mapper and Gait Fine-grained Detector to map and establish vocabulary relations into the gait space, effectively tackling modality heterogeneity and refining gait features.

• We evaluate $\alpha$-Gait on CASIA-B, CCPG, SUSTech1K, Gait3D and GREW, achieving superior performance and providing valuable insights.

## 2   Related Work

### 2.1   Gait Recognition

**Model-Based Gait Recognition.**  PoseGait [12] lifts 2D images to 3D poses and learns spatio-temporal cues with a multi-loss scheme for robustness. GaitGraph/GaitGraph2 [13, 23] use GCNs on 2D pose sequences to model motion while reducing appearance sensitivity. GaitTR [14] couples spatial transformers with temporal convolutions. GaitMixer [24] mixes spatial self-attention with large-kernel temporal convolutions. GPGait [15] improves generalization via a unified pose representation. SMPLGait [16] encodes shape and motion with dense 3D body models. SkeletonGait [17], HiH [25], and GaitHeat [18] use Gaussian-style maps to strengthen structural cues.

**Appearance-Based Gait Recognition.** GaitSet [2] views silhouettes as an unordered set. GaitPart [3] exploits part-wise signals. GaitGL [4] combines local and global 3D convolutions. GaitBase [5] is a simple, strong foundation for in-the-wild use. DANet [6], DyGait [26], HSTL [27], VPNet [7], GLGait [28], and GaitMoE [10] emphasize dynamic modeling.  GaitGCI [29], GaitCSV [19], CLTD [30], and GaitC$^3$I [31] apply causal inference to curb covariate effects. Origins [32] leverages generative diffusion to mitigate semantic inconsistency and uniformity. Beyond silhouettes, parsing-based inputs (GaitParsing [33], LandmarkGait [34], ParsingGait [35]) capture fine-grained parts. RGB pipelines (GaitEdge [36], BigGait [37]) enable end-to-end learning. point clouds (LidarGait [38]) address occlusion.  multi-modal designs (MMGaitFormer [39], CL-Gait [40]) enrich cues.  and Gait-X [41] builds an X-modality via patch-wise DCT for stronger in-/cross-domain performance.

### 2.2   Vision-Language Models

Gait-World derives its vocabulary space from a Text Encoder built on either Vision-Language Models (VLMs) or purely textual Large Language Models (LLMs).

**VLMs.** (i) **CLIP** [42]: contrastive image-text embeddings enabling broad zero-shot transfer and prompt-based retrieval/classification. (ii) **LLaVA** [43]: a CLIP-style visual encoder connected to an LLM via a lightweight adapter for instruction-following multimodality. (iii) **Qwen** [44]: Qwen-VL supports multi-image, high-resolution inputs with strong captioning and grounding for fine-grained semantics. (iv) **GPT** [45]: GPT-4V (and 4o) accepts images for VQA and multi-step reasoning over visual content.

**LLMs.** (i) **LLaMA** [46]: a widely used 7B-65B base family for downstream NLP adapters and tools. (ii) **GPT** [45]: GPT-3/4 exhibit in-context learning, instruction following, and strong general reasoning in text-only settings. (iii) **DeepSeek** [47]: V3/R1 emphasize efficiency and explicit reasoning with MoE and RL-style training, improving coding and mathematical tasks.

## 2.3 Vocabulary-Guided Learning

Using an explicit vocabulary links visual evidence to linguistic semantics. We outline two representative directions: Open-Vocabulary Learning and Text-Guided Learning.

**Open-Vocabulary Object Detection.** The goal is to detect objects beyond a fixed training label set by transferring language-aware knowledge. ViLD [48] distills a VLM teacher into a region-based detector to generalize to unseen categories. Detic [49] adds image-level supervision from large-scale VLM pretraining so one model handles both in- and out-of-vocabulary classes. OV-DETR [50] couples a transformer detector with vision-language pretraining, predicting categories directly from text embeddings.

**Text-Guided Face Recognition.** Text serves as guidance or supervision to refine identity features across granularities. CFAM [51] aligns images and captions at multiple resolutions. CaptionFace [52] combines a GPTFace component with a multi-scale feature alignment module. TGFR [53] uses cross-modal contrastive learning over global-local face-caption pairs.

**Discussion.** *Vocabulary-Guided Gait Recognition* instead uses vocabulary as priors to query identity-relevant gait cues. Unlike open-vocabulary detectors that treat words as labels, and text-guided face recognition where VLMs plug in directly, current VLMs are not yet sensitive to gait, requiring additional alignment.

## 3 Methodology

In this section, we first present the formulation of vocabulary-guided gait recognition in Sec. 3.1, then offer a comprehensive description of $\alpha$-Gait in Sec. 3.2, followed by the training and inference details in Sec. 3.3. Finally, we discuss various aspects of this work in Sec. 3.4.

### 3.1 Vocabulary-Guided Gait Recognition

We begin with the gait silhouette modality and appearance-based gait network for the simplicity and efficiency. A vanilla gait framework typically takes a silhouette sequence $\mathcal{X}$ as input, then extracts gait features using a Gait Encoder $\mathcal{E}$. Next, Horizontal Partitioning $\mathcal{P}$ is applied to obtain fine-grained gait part features $\mathcal{O}$, which are finally mapped to $\mathcal{F}$ for recognition through a Gait Head $\mathcal{G}$. This process can be as follows:

$$\mathcal{O} = \mathcal{P}(\mathcal{E}(\mathcal{X})) \tag{1}$$

$$\mathcal{F} = \mathcal{G}(\mathcal{O}) \tag{2}$$

where $\mathcal{X} \in \mathbb{R}^{\mathcal{S} \times \mathcal{H} \times \mathcal{W}}, \mathcal{O} \in \mathbb{R}^{\mathcal{C}_g \times \mathcal{P}}, \mathcal{F} \in \mathbb{R}^{\mathcal{C}_g \times \mathcal{P}}$, and $\mathcal{C}_g, \mathcal{S}, \mathcal{H}, \mathcal{W}, \mathcal{P}$ represent channel, consecutive $\mathcal{S}$ frames, height, width and the number of horizontal parts. This process relies on learning solely from gait modalities on limited gait databases, which makes it difficult to learn universal gait features.

To address this, we introduce Vocabulary-Guided Gait Recognition, dubbed Gait-World, which leverages vocabulary information from VLMs for better human understanding and gait semantic guidance. Gait-World comprises Vocabulary Base $\mathcal{V}$, VLMs $\mathcal{T}$, and Gait Network. Because VLMs are insufficiently sensitive to gait silhouette modality, we prepend the qualifier "human gait silhouette" to all vocabularies, which provides more precise descriptions for VLMs (*e.g.*, "human gait silhouette initial contact"). For convenience, this qualifier is omitted in subsequent discussions. Next, we provide more details for Gait-World, which mainly consists of three components:

**Vocabulary Base.** We predefine Vocabulary Base consisting of {"initial contact", "loading response", "mid stance", "terminal stance", "pre-swing", "initial swing", "mid swing", "terminal swing"}. As Figure 2 shows, the selection is motivated by the observation that VLMs are sensitive to the gait cycle actions that depend on the global details of gait silhouettes, but less sensitive to local details (*e.g.*, distinguishing legs from the silhouettes). Therefore, VLMs can accurately determine the phase of the gait cycle, which is crucial for gait recognition. In practice, Gait-World uses the Vocabulary Base to bridge the gap between the VLM and the gait spaces.

**VLMs.** Gait-World aims to harness the capacity of Vision-Language Models (*e.g.*, CLIP) or Large Language Models (*e.g.*, DeepSeek R1), which embody rich and universal knowledge. Specifically, the Text Encoder trained on large-scale public data develops a vocabulary space that generalizes well

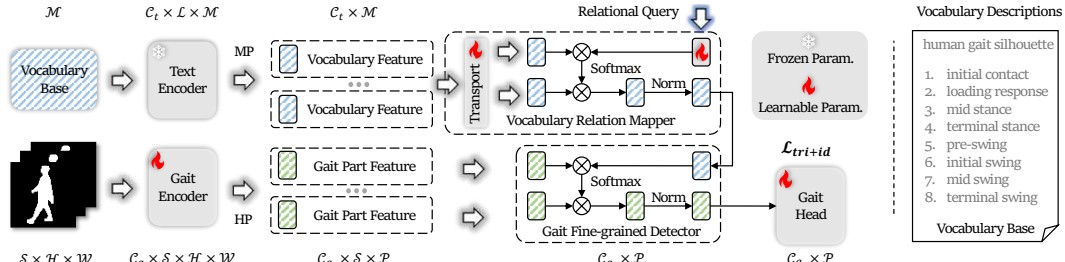

Figure 3: $\alpha$-Gait employs the vocabulary features from Text Encoder to guide the gait feature learning. HP denotes Horizontal Partition. Gait Encoder consists of several convolution blocks, and Gait Head includes Separate FCs and BNNeck. After the gait sequence passes through Gait Encoder and HP, the gait part features are fed into the Vocabulary Relation Mapper and Gait Fine-grained Detector, which guide them with vocabulary information for identification.

to real-world scenarios, which enables to guide the feature learning in the Gait Network. This process can be as follows:

$$\mathcal{V}_f = \mathcal{T}(\mathcal{V}) \tag{3}$$

where $\mathcal{V}_f \in \mathbb{R}^{\mathcal{C}_t \times \mathcal{L} \times \mathcal{M}}$, and $\mathcal{C}_t$, $\mathcal{L}$ and $\mathcal{M}$ denote channel, the number of tokens and the number of vocabulary. Note that the number of tokens assigned to each vocabulary feature can vary due to the differences in lexical length and tokenizer segmentation. For example, "pre-swing" and "initial contact" are tokenized into 1 and 2 tokens, respectively. Therefore, we employ the simple yet effective Mean Pooling (MP) to aggregate multiple tokens within a sentence into one token, *i.e.*, $\mathcal{V}_f \in \mathbb{R}^{\mathcal{C}_t \times \mathcal{M}}$.

**Gait Network.** The vocabulary features from VLMs serves as external information for guiding gait network learning. Therefore, the Gait-World paradigm is not constrained by current gait modalities or networks where vocabulary information solely guides gait representation learning.

## 3.2 Model Architecture

Towards Vocabulary-Guided Gait Recognition, we propose the first-generation model $\alpha$-Gait in this series, which primarily incorporates vocabulary information after the Gait Encoder and Horizontal Partitioning (*i.e.*, gait part features) due to the Vocabulary Base with temporal information and gait recognition with the fine-grained problem. Based on Gait-World, $\alpha$-Gait introduces the Vocabulary Relation Mapper (VRM) and the Gait Fine-grained Detector (GFD), addressing the discrepancies between the vocabulary and gait feature spaces, as well as the lack of universality of gait features.

**Vocabulary Instruction.** As shown in Figure 3, $\alpha$-Gait firstly obtains the vocabulary features $\mathcal{V}_f \in \mathbb{R}^{\mathcal{C}_t \times \mathcal{M}}$ from Text Encoder. *Note that gait cycle actions are an inherent attribute shared by all individuals and rely on the overall structure in silhouettes. Therefore, vocabulary features are shared across the body parts of all gait samples.*

$\alpha$-Gait extracts the gait part feature $\mathcal{O} \in \mathbb{R}^{\mathcal{C}_g \times \mathcal{S} \times \mathcal{P}}$ from Gait Encoder $\mathcal{E}$ and Horizontal Partitioning $\mathcal{P}$, preserving the temporal information for vocabulary guidance. *Note that each gait part feature contains the distinct walking pattern. Therefore, both Vocabulary Relation Mapper and Gait Fine-grained Detector are independent for each gait part feature, and we omit the part index for simplicity.* Given the vocabulary features $\mathcal{V}_f \in \mathbb{R}^{\mathcal{C}_t \times \mathcal{M}}$ and one gait part feature $\mathcal{O} \in \mathbb{R}^{\mathcal{C}_g \times \mathcal{S}}$, the VRM and GFD are as follows:

**Vocabulary Relation Mapper.** Although Gait-World carefully selects Vocabulary Base that is highly relevant to gait, there remains a significant feature distribution discrepancy between the Text Encoder and Gait Encoder. This discrepancy arises mainly for two reasons. Firstly, the Text Encoder modeling paradigm is based on sequential processing of the text modality. Secondly, the Text Encoder learning framework primarily relies on autoregressive Next-Token Prediction or Contrastive Learning from RGB Image-Text pairs. To address these issues, VRM firstly introduces Transition module to align vocabulary features into the gait space, and then Relational Query $\mathcal{Q}_R$ with the attention mechanism to establish associations among vocabularies, which enables the gait network to understand the

vocabularies in a more fine-grained manner. The process is as follows:

$$\mathcal{V}'_f = \text{ReLU}(\text{LN}(\text{Linear}(V_f))) \tag{4}$$

$$\mathcal{Q}_R = \mathcal{Q}_R, \quad \mathcal{K}_R = \mathcal{V}'_f, \quad \mathcal{V}_R = \mathcal{V}'_f \tag{5}$$

$$\mathcal{Q}_V = \text{Softmax}(\mathcal{Q}_R \otimes \mathcal{K}_R) \otimes \mathcal{V}_R \tag{6}$$

where $\text{Linear} \in \mathbb{R}^{\mathcal{C}_t \times \mathcal{C}_g}$, $\mathcal{Q}_R \in \mathbb{R}^{\mathcal{C}_g \times 1}$, $\mathcal{K}_R \in \mathbb{R}^{\mathcal{C}_g \times \mathcal{M}}$, $\mathcal{V}_R \in \mathbb{R}^{\mathcal{C}_g \times \mathcal{M}}$, $\mathcal{Q}_V \in \mathbb{R}^{\mathcal{C}_g \times 1}$. VRM eliminates the linear mapping in the attention mechanism to preserve the original vocabulary feature distribution as much as possible. Moreover, VRM normalizes the aligned vocabulary feature $\mathcal{Q}_V$ for guiding gait feature learning.

**Gait Fine-grained Detector.** Within the Gait-World paradigm, GFD primarily achieves that the vocabulary feature $\mathcal{Q}_V$ guides the gait feature learning with the corresponding semantic information for the complex real-world scenarios. To this end, GFD treats the process from an object detection perspective where the vocabulary feature detects gait features with the corresponding semantics. Similar to the DETR [54], GFD presents $\mathcal{Q}_V$ as the object query and the gait part feature $\mathcal{O}$ as the regions of interest. The process is as follows:

$$\mathcal{Q}_V = \mathcal{Q}_V, \quad \mathcal{K}_V = \mathcal{O}, \quad \mathcal{V}_V = \mathcal{O} \tag{7}$$

$$\mathcal{F} = \text{Softmax}(\mathcal{Q}_V \otimes \mathcal{K}_V) \otimes \mathcal{V}_V \tag{8}$$

where $\mathcal{Q}_V \in \mathbb{R}^{\mathcal{C}_g \times 1}$, $\mathcal{K}_V \in \mathbb{R}^{\mathcal{C}_g \times \mathcal{S}}$, $\mathcal{V}_V \in \mathbb{R}^{\mathcal{C}_g \times \mathcal{S}}$, $\mathcal{F} \in \mathbb{R}^{\mathcal{C}_g \times 1}$. GFD also normalizes the detected gait feature $\mathcal{F}$ for the following Gait Head, easing the training process.

### 3.3  Training Details

**Training Stage.** $\alpha$-Gait aims to recognize the individual identity where vocabulary information from VLMs solely guides the gait feature extraction. Consequently, $\alpha$-Gait remains consistent with conventional gait recognition, including two types of identity loss. Triplet Loss [55] $\mathcal{L}_{tp}$ and Cross Entropy Loss $\mathcal{L}_{ce}$, constraining each part independently.

$$\mathcal{L} = \mathcal{L}_{tp} + \mathcal{L}_{ce} \tag{9}$$

**Inference Stage.** After training, a strong relation is established between the Vocabulary Base space from VLMs and the gait space. At the inference stage, $\alpha$-Gait remains the Vocabulary Base to refine gait features for real-world scenarios.

### 3.4  Discussion

To facilitate a clear grasp and significance of Vocabulary-Guided Gait Recognition, we further explain and clarify Gait-World, $\alpha$-Gait and the scope of this work:

**Gait-World** aims to provide a better human understanding of gaits, and a new paradigm to complement existing paradigms (*i.e.*, appearance-based and model-based methods). It serves as an intuitive and efficient tool for researchers to refine their understanding of gait patterns, providing new directions for gait research. Researchers only need to design better vocabulary prompts and share their embeddings with the gait community, without the burden of computational and memory overhead introduced by large models or the need for complex gait model architectures.

**$\alpha$-Gait** serves as an initial attempt, which aims to provide an intuitive demonstration of the vocabulary's effectiveness for gait recognition. Hence, we propose a simple yet effective architecture, rather than relying on complex frameworks for incremental performance gains.

**The relationships with multimodals.** Multimodal approaches typically require each input to be paired with the corresponding several modalities, which, despite offering information gains, also introduces challenges in data collection and computational overhead. $\alpha$-Gait serves as an initial attempt with only eight universal vocabulary embeddings shared across all inputs, and admittedly does not yet constitute a multimodal paradigm.

## 4  Experiments

The mainstream public gait databases and the implementation details are shown in Appendix A and the evaluations on our method are introduced in the next sections.

Table 1: The evaluation on CCPG with clothing-changing conditions.

| Paradigm | Method | Venue | Gait Evaluation Protocol | | | | | ReID Evaluation Protocol | | | | |
|---|---|---|---|---|---|---|---|---|---|---|---|---|
| | | | CL | UP | DN | BG | Mean | CL | UP | DN | BG | Mean |
| Model | GaitGraph2 [13] | CVPRW22 | 5.0 | 5.3 | 5.8 | 6.2 | 5.1 | 5.0 | 5.7 | 7.3 | 8.8 | 6.7 |
| | Gait-TR [14] | ES23 | 15.7 | 18.3 | 18.5 | 17.5 | 17.5 | 24.3 | 28.7 | 31.1 | 28.1 | 28.1 |
| | MSGG [56] | MTA23 | 29.0 | 34.5 | 37.1 | 33.3 | 33.5 | 43.1 | 52.9 | 57.4 | 49.9 | 50.8 |
| | SkeletonGait [17] | AAAI24 | 40.4 | 48.5 | 53.0 | 61.7 | 50.9 | 52.4 | 65.4 | 72.8 | 80.9 | 67.9 |
| Appearance | GaitSet [2] | AAAI19 | 60.2 | 65.2 | 65.1 | 68.5 | 64.8 | 77.5 | 85.0 | 82.9 | 87.5 | 83.2 |
| | GaitPart [3] | CVPR20 | 64.3 | 67.8 | 68.6 | 71.7 | 68.1 | 79.2 | 85.3 | 86.5 | 88.0 | 84.8 |
| | OGBase [57] | CVPR23 | 52.1 | 57.3 | 60.1 | 63.3 | 58.2 | 70.2 | 76.9 | 80.4 | 83.4 | 77.7 |
| | GaitBase [5] | CVPR23 | 71.6 | 75.0 | 76.8 | 78.6 | 75.5 | 88.5 | 92.7 | 93.4 | 93.2 | 92.0 |
| | DeepGaitV2 [58] | TPAMI25 | 78.6 | 84.8 | 80.7 | 89.2 | 83.3 | 90.5 | 96.3 | 91.4 | 96.7 | 93.7 |
| Gait-World | $\alpha$-Gait-S (**ours**) | NeurIPS25 | **82.8** | **89.0** | **84.6** | **92.7** | **87.3** | **92.0** | **98.1** | **93.4** | **96.9** | **95.1** |

Table 2: The evaluation on SUSTech1K with different attributes (abbrev.: NM=Normal, BG=Bag, CL=Clothing, CRY=Carrying, UMB=Umbrella, UNI=Uniform, OCC=Occlusion, NT=Night).

| Paradigm | Method | Venue | Probe Sequence | | | | | | | | Overall | |
|---|---|---|---|---|---|---|---|---|---|---|---|---|
| | | | NM | BG | CL | CRY | UMB | UNI | OCC | NT | Rank-1 | Rank-5 |
| Model | GaitGraph2 [13] | CVPRW22 | 22.2 | 18.2 | 6.8 | 18.6 | 13.4 | 19.2 | 27.3 | 16.4 | 18.6 | 40.2 |
| | Gait-TR [14] | ES23 | 33.3 | 31.5 | 21.0 | 30.4 | 22.7 | 34.6 | 44.9 | 23.5 | 30.8 | 56.0 |
| | MSGG [56] | MTA23 | 67.1 | 66.2 | 35.9 | 63.3 | 61.6 | 58.1 | 66.6 | 17.9 | 33.8 | - |
| | SkeletonGait [17] | AAAI24 | 67.9 | 63.5 | 36.5 | 61.6 | 58.1 | 67.2 | 79.1 | 50.1 | 63.0 | 83.5 |
| Appearance | GaitSet [2] | AAAI19 | 69.1 | 68.2 | 37.4 | 65.0 | 63.1 | 61.0 | 67.2 | 23.0 | 65.0 | 84.8 |
| | GaitPart [3] | CVPR20 | 62.2 | 62.8 | 33.1 | 59.5 | 57.2 | 54.8 | 57.2 | 21.7 | 59.2 | 80.8 |
| | GaitGL [4] | ICCV21 | 67.1 | 66.2 | 35.9 | 63.3 | 61.6 | 58.1 | 66.6 | 17.9 | 63.1 | 82.8 |
| | GaitBase [5] | CVPR23 | 81.5 | 77.5 | 49.6 | 75.8 | 75.5 | 76.7 | 81.4 | 25.9 | 76.1 | 89.4 |
| | DeepGaitV2 [58] | TPAMI25 | 87.4 | 84.1 | 53.4 | 81.3 | 86.1 | 84.8 | 88.5 | 28.8 | 82.3 | 92.5 |
| Gait-World | $\alpha$-Gait-S (**ours**) | NeurIPS25 | **91.1** | **87.2** | **64.0** | **85.3** | **89.5** | **88.8** | **92.7** | 28.2 | **86.3** | **93.9** |

## 4.1 Results on Constrained Scenario

**CASIA-B.** As shown in Table 3, $\alpha$-Gait-T achieves competitive performance under all conditions, with an average accuracy of 94.8%, proving the universality of gait cycle action under NM, BG, and CL scenarios. Specifically, $\alpha$-Gait-T approaches the SOTA on NM (98.9 %) and BG (96.8%).

**CCPG.** As shown in Table 1, $\alpha$-Gait-S significantly outperforms appearance-based and model-based methods in the more challenging full-body clothing change scenarios. For instance, it exceeds DeepGaitV2 by 4% in mean accuracy, demonstrating that textual information can better guide the model in learning covariate-independent features.

## 4.2 Results on In-the-wild Scenario

**SUSTech1K.** In real-world scenarios, such as occlusions, umbrella usage, and varying lighting conditions, $\alpha$-Gait-T significantly surpasses previous SOTA methods shown in Table 2. For example, it outperforms DeepGaitV2 by 4% in Rank-1 accuracy, demonstrating the feature robustness of the text-guided gait cycle actions.

**Gait3D.** In larger-scale scenarios, although silhouette-based methods are approaching saturation due to the impact of covariates on upstream segmentation algorithms, $\alpha$-Gait-M is still able to improve performance shown in Table 4. For instance, it surpasses GaitMoE by 2.6% in Rank-1 accuracy, indicating that the Gait-World paradigm can serve as a valuable complement to existing approaches.

**GREW.** Similarly, GREW is also significantly affected by upstream gait modal extraction algorithms, with silhouette-based methods approaching the limitations. As shown in Table 4, $\alpha$-Gait-L achieves competitive results, surpassing VPNet [7] by 1.2%. $\alpha$-Gait-L adopts Free Lunch [61] (*i.e.*, logits as gait features) to achieve more stable results without introducing additional computational complexity.

Table 3: The evaluation on CASIA-B under different conditions with Rank-1 accuracy (%).

| Paradigm | Method | Venue | NM | BG | CL | Mean |
|---|---|---|---|---|---|---|
| Model | GaitGraph2 [23] | CVPRW22 | 80.3 | 71.4 | 63.8 | 71.8 |
| | GaitTR [14] | ES23 | 94.7 | 89.3 | 86.7 | 90.2 |
| | GPGait [15] | ICCV23 | 93.6 | 80.2 | 69.3 | 81.0 |
| Appearance | GaitSet [2] | AAAI19 | 95.0 | 87.2 | 70.4 | 84.2 |
| | GaitPart [3] | CVPR20 | 96.2 | 91.5 | 78.7 | 88.8 |
| | GLN [59] | ECCV20 | 96.9 | 94.0 | 77.5 | 89.5 |
| | GaitGL [4] | ICCV21 | 97.4 | 94.5 | 83.6 | 91.8 |
| | QAGait [60] | AAAI24 | 97.9 | 94.6 | 78.2 | 90.2 |
| | GaitBase [5] | CVPR23 | 97.6 | 94.0 | 77.4 | 89.8 |
| | DANet [6] | CVPR23 | 98.0 | 95.9 | 89.9 | 94.6 |
| | GaitGCI [29] | CVPR23 | 97.9 | 95.0 | 86.4 | 93.1 |
| | DyGait [26] | ICCV23 | 98.4 | 96.2 | 87.8 | 94.1 |
| | HSTL [27] | ICCV23 | 98.1 | 95.9 | 88.9 | 94.3 |
| | VPNet [7] | CVPR24 | 98.3 | 96.3 | 90.0 | 94.9 |
| | DeepGaitV2 [58] | TPAMI25 | – | – | – | 89.6 |
| | CLTD [30] | ECCV24 | 98.6 | 96.4 | 89.3 | 94.8 |
| | Free Lunch [61] | ECCV24 | 98.1 | 94.1 | 77.9 | 90.0 |
| Gait-World | $\alpha$-Gait-T (**ours**) | NeurIPS25 | **98.9** | **96.8** | **88.6** | **94.8** |

Table 4: The evaluation on Gait3D and GREW.

| Paradigm | Method | Venue | Gait3D | | | GREW | | |
|---|---|---|---|---|---|---|---|---|
| | | | Rank-1 | Rank-5 | mAP | Rank-1 | Rank-5 | Rank-10 |
| Model | GaitGraph2 [23] | CVPRW22 | 11.2 | - | - | 64.8 | - | - |
| | GaitTR [14] | ES23 | 7.2 | - | - | 48.6 | - | - |
| | GPGait [15] | ICCV23 | 22.4 | - | - | 57.0 | - | - |
| Appearance | GaitSet [2] | AAAI19 | 36.7 | 58.3 | 30.0 | 46.3 | 63.6 | 70.3 |
| | GaitPart [3] | CVPR20 | 28.2 | 47.6 | 47.6 | 44.0 | 60.7 | 67.3 |
| | GaitGL [4] | ICCV21 | 29.7 | 48.5 | 22.3 | 47.3 | 63.6 | – |
| | MTSGait [62] | MM22 | 48.7 | 67.1 | 37.6 | 55.3 | 71.3 | 76.9 |
| | QAGait [60] | AAAI24 | 67.0 | 81.5 | 56.5 | 59.1 | 74.0 | 79.2 |
| | GaitBase [5] | CVPR23 | 64.6 | – | – | 60.1 | – | – |
| | GaitGCI [29] | CVPR23 | 50.3 | 68.5 | 39.5 | 68.5 | 80.8 | 84.9 |
| | DyGait [26] | ICCV23 | 66.3 | 80.8 | 56.4 | 71.4 | 83.2 | 86.8 |
| | HSTL [27] | ICCV23 | 61.3 | 76.3 | 55.5 | 62.7 | 76.6 | 81.3 |
| | VPNet [7] | CVPR24 | 75.4 | 87.1 | – | 80.0 | 89.4 | – |
| | DeepGaitV2 [58] | TPAMI25 | 74.4 | 88.0 | 65.8 | 77.7 | 88.9 | 91.8 |
| | CLTD [30] | ECCV24 | 69.7 | 85.2 | – | 78.0 | 87.8 | – |
| | GaitMoE [10] | ECCV24 | 73.7 | – | 66.2 | 79.6 | 89.1 | – |
| | Free Lunch [61] | ECCV24 | 70.1 | – | 61.9 | 65.5 | 78.7 | 83.3 |
| Gait-World | $\alpha$-Gait-M/L (**ours**) | NeurIPS25 | **76.3** | **87.7** | **67.8** | **81.2** | **90.2** | **92.7** |

## 4.3 Ablation Study

In this section, we validate the universality of the Gait-World with different Text Encoder, and illustrate the modality and vocabulary expansions. Additionally, we visualize the mechanism of vocabulary guidance, and analyze the trade-off of $\alpha$-Gait between accuracy and efficiency.

**The effectiveness of Gait-World.** As shown in Table 5, although Gait Network relying solely on gait silhouettes and adaptive learning achieves competitive results, under the Gait-World paradigm, $\alpha$-Gait further improves Rank-1 accuracy by 2.6% on Gait3D. This indicates that the vocabulary space distribution derived from Large Language Models possesses greater universality.

Table 5: The ablation study on Gait3D and CCPG.

| Method | Gait3D | | CCPG | | | | |
|---|---|---|---|---|---|---|---|
| | Rank-1 | mAP | CL | UP | DN | BG | Mean |
| $\alpha$-Gait | 76.2 | 67.8 | 82.8 | 89.0 | 84.6 | 92.7 | 87.3 |
| Gait Encoder | 73.6 | 65.1 | 80.1 | 86.4 | 83.9 | 91.4 | 85.5 |
| The analysis on Text Encoder | | | | | | | |
| Initial Random | 71.8 | 63.3 | 77.7 | 84.2 | 77.6 | 86.2 | 81.4 |
| Learnable Query | 74.1 | 66.8 | 82.3 | 89.0 | 83.9 | 92.5 | 86.9 |
| CLIP | 75.2 | 67.7 | 82.2 | 88.9 | 85.0 | 92.8 | 87.2 |
| LlaMa | 74.8 | 66.9 | 82.5 | 89.3 | 84.3 | 93.0 | 87.3 |
| DeepSeekR1-Distill | 76.2 | 67.8 | 82.8 | 89.0 | 84.6 | 92.7 | 87.3 |

Table 6: Ablations on modality and vocabulary expansions on Gait3D

| Paradigm | Variant | Vocabulary Base | Rank-1 | mAP |
|---|---|---|---|---|
| Appearance | Gait Encoder | — | 73.6 | 65.1 |
| | Gait Encoder w/ $\alpha$-Gait | [8 phases] | **76.2** | **67.8** |
| Model | SkeletonGait | — | 37.9 | 29.6 |
| | SkeletonGait w/ $\alpha$-Gait | [8 phases] | **40.3** | **31.3** |
| Multimodel | SkeletonGait++ | — | 76.0 | 69.2 |
| | SkeletonGait++ w/ $\alpha$-Gait | [8 phases] | **78.1** | **71.5** |
| Gait-World | $\alpha$-Gait | [8 phases] | 76.3 | 67.8 |
| | $\alpha$-Gait w/ More vocabularies | [8 phases + view, bag, clothing inv.] | **76.8** | **68.4** |

**The analysis on Text Encoder.** To analyze the importance of vocabulary features, we first replace them with Initial Random features. As shown in Table 5, the results show that non-relational features disrupt gait learning, thereby validating the effectiveness of $\alpha$-Gait from the benefits of strongly associated vocabulary features rather than the architectures (*i.e.*, VRM and GFD). Additionally, we replace with Learnable Query, which starts from normal distribution but adapts through VRM and GFD to learn relevant information from the gait features, confirming the architecture's efficiency. Furthermore, we substituted different Text Encoders, leading to different improvements. DeepSeek-R1-Distill, with its universal vocabulary reasoning capabilities, produced superior vocabulary features, while CLIP, leveraging image-text pairs for alignment, excelled in capturing visual features.

**The modality and vocabulary expansions.** As shown in Table 6, vocabulary guidance brings consistent gains. For the Appearance-based method, adding $\alpha$-Gait lifts Rank-1/mAP from 73.6/65.1 to 76.2/67.8 (+2.6/+2.7), indicating that phase-aware cues complement generic silhouette features. For Model-based method, SkeletonGait improves from 37.9/29.6 to 40.3/31.3 (+2.4/+1.7), showing larger benefits when the baseline is weaker. In Multimodel-based method, SkeletonGait++ also increases from 76.0/69.2 to 78.1/71.5 (+2.1/+2.3), meaning the guidance remains effective with stronger backbones. Within Gait-World, expanding beyond the eight phases with view-angle, bag, and clothing invariants brings a further rise from 76.3/67.8 to 76.8/68.4 (+0.5/+0.6), consistent with the goal of suppressing appearance confounders.

**The visualization of vocabulary guidance.** We provide qualitative analysis to validate that $\alpha$-Gait extracts gait features related to vocabulary information and provide meaningful feedback for humans. As shown in Figure 4, given the eight gait cycle vocabularies, we visualize the silhouettes with the highest Softmax response in the attention mechanism. It can be observed that GFD accurately captures gait cycle actions under various covariates, revealing that the $\alpha$-Gait indeed understands the human vocabulary. Meanwhile, it also inspires researchers to better understand gait.

**The efficiency-accuracy trade-off.** As shown in Figure 5, $\alpha$-Gait (59.1 M params, 85.6 G FLOPs) reaches 76.3% accuracy, sitting on the frontier of this cohort. Versus DeepGaitV2 (25.5 M / 85.3 G / 74.4%), it uses nearly the same compute (+0.3 G FLOPs) yet improves accuracy by 1.9% by steering attention to phase-specific, identity-bearing cues via vocabulary-guided detection. Relative to DyGait (133.1 M / 239.0 G / 66.3%), it is lighter by 74.0 M parameters and 153.4 G FLOPs while improving

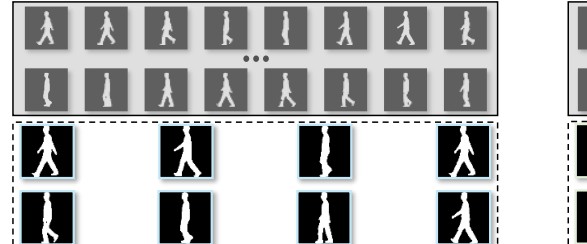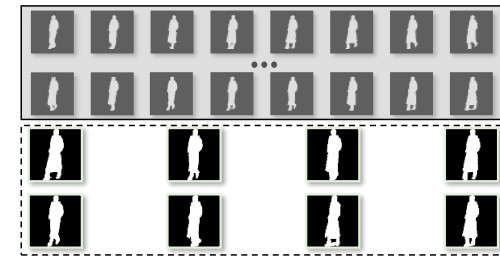

Figure 4: The gray box represents the complete gait sequence. Give the eight gait cycle vocabularies, GFD detects the eight silhouettes with the highest Softmax response in the attention mechanism, shown in the dash boxes.

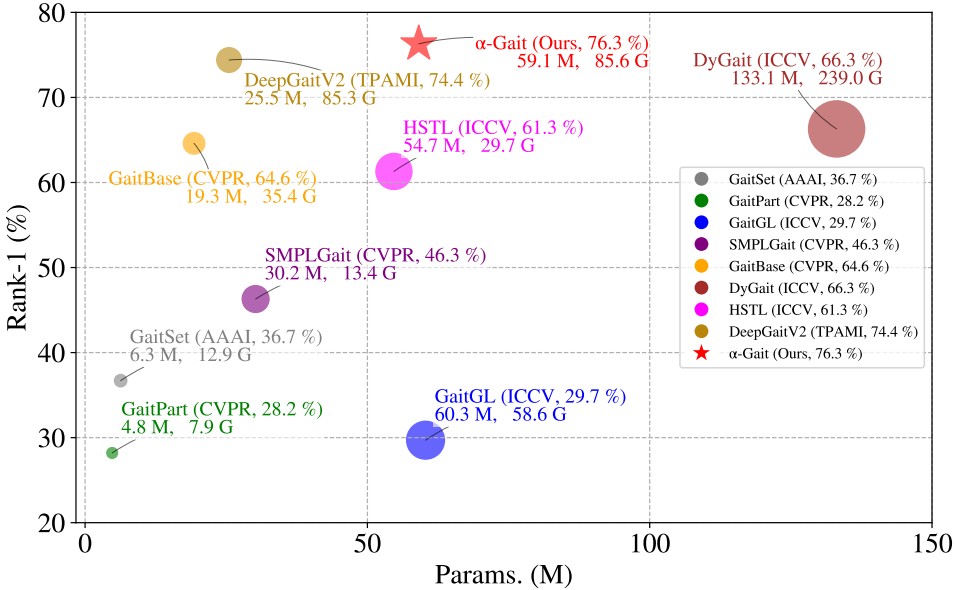

Figure 5: Model comparison on Gait3D: Rank-1 (%) vs. parameters (M) and FLOPs (G).

accuracy by 10.0%, as VRM-based alignment injects human-understandable gait terms that suppress clothing/view confounders and reduce the search space. At training and inference stages, the text encoder is frozen and word embeddings are cached offline, adding negligible runtime overhead.

## 5  Conclusion and Limitations

In this work, we introduce the vocabulary to the gait field due to the inherent interpretability and semantic guidance. Specifically, we propose a novel paradigm Gait-World, which aims to explore gait concepts with human vocabulary and VLMs. Gait-World integrates vocabulary information into the Gait Network by leveraging gait cycle action vocabularies, thereby enhancing human understanding of gaits. Furthermore, we introduce $\alpha$-Gait, the first model under the Gait-World paradigm, which utilizes VRM and GFD to more precisely guide gait feature learning with corresponding vocabulary features. Extensive experiments on multiple complex gait databases prove the universality.

**Limitations and Future Works.** $\alpha$-Gait serves as an initial attempt with eight universal vocabulary embeddings preliminarily validates the value of vocabulary information for gait recognition, whereas the more comprehensive exploitation of vocabulary information yields richer benefits, such as gait attribute learning with the vocabulary labels. In future work, it can be extended to be a multimodal paradigm, providing each input with a unique language description, enabling richer gait features. Additionally, a detailed discussion of risks and safeguards is provided in Appendix B. In conclusion, Gait-World provides a better human understanding of gaits, and a new paradigm to complement existing paradigms.

# Acknowledgement

This work is jointly supported by National Natural Science Foundation of China (62276025, 62206022, 62476027) and the Fundamental Research Funds for the Central Universities (2253200026).

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

# A Databases and Implementation Details

Table 7: Id. and Seq. denote the number of identities and sequences. CV, BG and CL refer to cross-view and carrying bags and cross-clothing conditions. $\mathcal{D}$ and $\mathcal{C}$ denote the number of conv blocks and the channels in each visual stage.

| Environment | Dataset | Train | | Test | | Condition | Stage | Channels | Strides |
|---|---|---|---|---|---|---|---|---|---|
| | | Id. | Seq. | Id. | Seq. | | $[\mathcal{D}_1, \mathcal{D}_2, \mathcal{D}_3, \mathcal{D}_4]$ | $[\mathcal{C}_1, \mathcal{C}_2, \mathcal{C}_3, \mathcal{C}_4]$ | |
| Constrained | CASIA-B [63] | 74 | 8,140 | 50 | 5,500 | CV, BG, CL | [1, 1, 1, -] | [64, 128, 256, -] | [1, 2, 1, -] |
| | CCPG [57] | 100 | 8,187 | 100 | 8,095 | CV, BG, CL | [1, 1, 1, 1] | [64, 128, 256, 512] | [1, 2, 2, 1] |
| In-the-wild | SUSTech1K [38] | 200 | 5988 | 850 | 19,228 | Real-world | [1, 1, 1, 1] | [64, 128, 256, 512] | [1, 2, 2, 1] |
| | Gait3D [16] | 3,000 | 18,940 | 1,000 | 6,369 | Real-world | [1, 4, 4, 1] | [64, 128, 256, 512] | [1, 2, 2, 1] |
| | GREW [64] | 20,000 | 102,887 | 6,000 | 24,000 | Real-world | [2, 4, 4, 2] | [64, 128, 256, 512] | [1, 2, 2, 1] |

## A.1 Databases

Gait databases are commonly categorized into two groups: Constrained and In-the-wild scenarios. As shown in Table 7, CASIA-B[63], CCPG [57] generally include fewer individuals but provide explicit condition types. In-the-wild databases SUSTech1K [38], Gait3D [16] and GREW [64] contain a larger number of identities and more challenging scenarios (*e.g.*, occlusions).

**CASIA-B** [63] includes 124 subjects recorded from 11 view angles, which contains Normal Walking (NM), Carrying Bags (BG) and Clothing-Changing (CL) conditions.

**CCPG** [57] concentrates on the effects of clothing variations, including 200 individuals with more than 16,000 sequences. By providing fine-grained clothing variations and realistic challenges, CCPG helps researchers investigate how to handle cloth-changing issues more effectively.

**SUSTech1K** [38] is a large-scale, multimodal gait dataset collected by a LiDAR sensor and an RGB camera. It comprises 1,050 subjects, including diverse real-world conditions (*e.g.*, clothing, night-time, and view angles scenarios).

**Gait3D** [16] is a large-scale gait database collected from 39 cameras in a supermarket with factors like occlusions and view angles. It includes 3,000 subjects, divided into a training subset of 2,000 and a testing subset of 1,000.

**GREW** [64] is a large-scale in-the-wild database comprising 26,345 subjects and 128,671 sequences collected from 882 cameras. Each sequence provides rich modalities, silhouettes, optical flow, and 2D/3D pose, enabling both appearance-based and model-based gait studies. The 20,000 subjects are designated for training and 6,000 for testing, and each test subject contributes two gallery sequences and two probe sequences.

## A.2 Implementation Details

We describe the training process below in detail:

**Inputs.** The silhouettes on all databases are transformed into $64 \times 44$, and each sequence consists of 30 consecutive frames. We adopt the mini-batch $[\mathcal{I}, \mathcal{J}]$ is consistent with [5], and $\mathcal{I}, \mathcal{J}$ denote the number of subjects and the number of sequences, respectively.

**Networks.** We provide four model types: $\alpha$-Gait-T, $\alpha$-Gait-S, $\alpha$-Gait-M, $\alpha$-Gait-L, improving the optimization on different-scale databases. All models employ the Stem module and 2D ResBlock in the first Stage, which is consistent with DeepGaitV2[58]. $\alpha$-Gait-T consists of 3 Stages with block numbers [1, 1, 1], channels [64, 128, 256], where the Bottleneck blocks place in the last 2 Stages. $\alpha$-Gait-S consists of 4 Stages with block numbers [1, 1, 1, 1], channels [64, 128, 256, 512], where the Bottleneck blocks place in the last 3 Stages. $\alpha$-Gait-M consists of 4 Stages with block numbers [1, 4, 4, 1], channels [64, 128, 256, 512], where the P3D blocks place in the last 3 Stages. $\alpha$-Gait-L consists of 4 Stages with block numbers [2, 4, 4, 2], channels [64, 128, 256, 512], where the P3D blocks place in the last 3 Stages.

**Optimization.** We employ SGD with an initial learning rate of 0.1, which is reduced by 0.1 at specific iteration milestones where CASIA-B, CCPG, SUSTech1K, Gait3D and GREW are $[20K, 40K, 50K]$, $[20K, 40K, 50K]$, $[20K, 30K, 40K]$, $[20K, 40K, 50K]$ and $[80K, 120K, 150K]$, respectively. The total training iterations of CASIA-B, CCPG, SUSTech1K, Gait3D and GREW are $60K$, $60K$,

$50K$, $60K$, $180K$, respectively. **Text Encoder.** We select CLIP, LlaMa3-8B, and DeepSeek-R1-Distill-Llama-8B as representative Vision-Language Models (VLMs) to validate the effectiveness of Gait-World.

## B  Responsible Use, Risks, and Safeguards

**Scope.** This work studies vocabulary-guided gait recognition under a research-only setting. All experiments use public datasets approved for academic use and silhouette/skeleton representations (no RGB/audio).

**Risks.** (1) Covert or indiscriminate surveillance; (2) use without informed consent; (3) unfair errors across sub-populations; (4) function creep beyond the stated research scope.

**Technical safeguards.**

- Release silhouettes/skeletons only; prohibit identity recovery and real-time CCTV deployment without explicit, informed consent.
- Freeze the text encoder and cache vocabulary embeddings at inference, keeping guidance overhead minimal and auditable.
- Provide subgroup reporting (e.g., gender/age/assistive devices); if disparity exceeds a preset threshold, retrain with re-weighting/fairness regularizers and document the outcome in the model card.

**Process and access controls.**

- Non-commercial, research-only license forbidding surveillance use; usage must document consent or a clear legal mandate.
- Gated access with per-request approval; logged usage; immediate revocation and public disclosure upon policy breach.
- Incident response: if any identity is recoverable, notify within 48 h and irreversibly delete the recovered data/models.

**Scope limitation.** The system is intended for academic benchmarking and analysis; deployment in operational surveillance or identification systems is out of scope.

