# OpenReview forum: "Vocabulary-Guided Gait Recognition"
_NeurIPS.cc/2025/Conference — NeurIPS 2025 poster_

### Official Review · Reviewer_dYxT · 2025-06-09

**Clarity:** 4
**Significance:** 4
**Originality:** 3
**Rating:** 5
**Confidence:** 3

**Summary:**

This paper proposes Gait-World, a new paradigm for gait recognition that uses human-interpretable vocabulary and Vision-Language Models to guide feature learning. And proposed α-Gait, which bridges text and gait modalities via a Vocabulary Relation Mapper and Fine-grained Detector. Experiments on multiple datasets show improved robustness to clothing changes and occlusions, demonstrating how semantic vocabulary can enhance gait recognition while maintaining efficiency.

**Questions:**

(1) In the problem description on lines 34-36, I think "Ambiguous gait concepts" would be a very appealing point, but the author didn't explain it clearly.  The same goes for the analysis in the abstract.

(2) The paper claims that VLMs are insensitive to gait silhouettes, but only the detection results of GPT are shown in Figure 2. Can the detection results of other VLMs be provided to support this view? And can all VLMs identify the gait cycle from the silhouettes?

(3) The predefined gait cycle actions appear biomechanically inspired. What empirical evidence supports that these 8 phases are optimal? Could a data-driven vocabulary discovery approach (e.g., clustering VLM embeddings) yield more discriminative terms? How does performance vary with vocabulary size/granularity?

(4) In line 195,  the author points out that "Horizontal Partitioning preserves temporal dimensions", but VRM/GFD processes each frame independently.

**Ethical Concerns:**

["NO or VERY MINOR ethics concerns only"]

**Final Justification:**

Thank you for your rebuttal. You've resolved most of my doubts. Thus I will maintain my rating.

**Limitations:**

Yes.

**Paper Formatting Concerns:**

There is no.

**Quality:**

4

**Strengths And Weaknesses:**

Strengths:

(1) The illustrations in this paper are beautiful and can convey the meaning of the text.

(2) The motivation of this paper is significant. It is quite convincing to utilize the universal features of VLMs to assist gait recognition.

(3) The writing of this article is easy to understand, and the language is fluent.

(4) The experimental results were extremely impressive.

Weaknesses:  Please refer to Limitations.

---

> ### Author Rebuttal · Authors · 2025-07-31
>
> Dear Reviewer dYxT:
>
> Thank you for the constructive critique and encouraging remarks. We respond to each concern in the following sections, and we welcome any follow-up questions you may have.
>
> ---
>
> > [W1] Please refer to Limitations.
>
> [A] Thank you for your valuable comment. The manuscript mainly provides limitations as follows:
>
> - ***(i)*** Limitation 1: Expanding the vocabulary. We try to explore the vocabularies that describe common covariates: “view-angle invariant”, “bag invariant”, “clothing invariant” and evaluate the impact as follows:
>
>     | Model | Vocabulary Base | Rank-1 (%) | mAP (%) |
>     |------|------|------|------|
>     |Baseline | -   | 74.4   |65.8 |
>     | $\alpha$-Gait   | [Initial contact, ..., loading response]   | 76.3   |67.8|
>     | $\alpha$-Gait   | [Initial contact, ..., loading response, view-angle invariant, bag invariant, clothing invariant]   | 76.8   |68.4|
>
>     As the table shows, we further explore how gait-related terms can potentially affect recognition performance and refine our understanding of gaits. With the flexibility and extensibility of Gait-World, we will continue to explore the vocabularies in future work, thereby deepening understanding within the gait domain.
>
> - ***(ii)*** Limitation 2: Extending the method to a multimodal paradigm. We extend α‑Gait from silhouettes to skeleton sequences, and the results confirm that the vocabulary guidance remains effective beyond a single modality. Specifically, we conduct the experiment with SkeletonGait[1] in α-Gait framework on Gait3D：
>
>     | Model | Venue | Rank-1 (%) | mAP (%) |
>     |------|------|------|------|
>     |SkeletonGait | AAAI24  | 37.9   |29.6 |
>     | SkeletonGait with α-Gait   | -  | 40.3   |31.3|
>     |SkeletonGait++ | AAAI24  | 76.0   |69.2 |
>     | SkeletonGait++ with α-Gait   | -  | 78.1   |71.5|
>
>     The table shows the reproduction performance where the vocabulary can also be effective in other modalities (i.e., Skeleton maps), which further proves the modality-generalizability.
>
>     [1] Fan C, Ma J, Jin D, et al. Skeletongait: Gait recognition using skeleton maps[C]//Proceedings of the AAAI conference on artificial intelligence. 2024.
>
> ---
>
> > [Q1] In the problem description on lines 34-36, I think "Ambiguous gait concepts" would be a very appealing point, but the author didn't explain it clearly. The same goes for the analysis in the abstract.
>
> [A] Thanks for this valuable suggestion. “Ambiguous gait concepts” indicate the broad gait terms, such as spatial, temporal, or structural information, that many studies use to characterize gait features without specifying exactly what they capture. Clarifying these concepts is crucial for two reasons:
>
> - ***(i)*** Slow Gait development. Vague terminology slows progress in the rapidly evolving gait recognition field. For example, research may only stop at designing better backbones instead of probing deeper into gait semantics.
>
> - ***(ii)*** Difficult practice verification. When new and precise gait concepts emerge, researchers often lack a straightforward way to implement and validate them.
>
> Gait-World serves as an intuitive and efficient tool for researchers to refine their understanding of gait patterns, and α-Gait further provides a demonstration of the vocabulary’s effectiveness.
>
> ---
>
> > [Q2] The paper claims that VLMs are insensitive to gait silhouettes, but only the detection results of GPT are shown in Figure 2. Can the detection results of other VLMs be provided to support this view? And can all VLMs identify the gait cycle from the silhouettes?
>
> [A] Thanks for raising this point. Owing to the latest NeurIPS rebuttal constraints (No permission for the additional images or links), we will present more results in the revised manuscript. We present the same silhouette and question (Figure 2 in the main manuscript) to three top-tier VLMs: GPT‑o3, Qwen3‑235B, and Kimi‑K2, which obtain highly consistent outcomes:
>
> - ***(i)*** GPT‑o3. Answer: (E) pre‑swing, Reason: The rear foot shows heel‑off with only the toes likely still touching, and the limb has started knee flexion and unloading—classic signs that it is just about to leave the ground to enter initial swing.
>
> - ***(ii)*** Qwen3‑235B: Answer: E, Reason: The stance leg (right) remains flat on the ground, and the swing leg (left) appears to initiate lifting with minimal knee flexion, aligning with pre-swing characteristics where the foot is still in contact.
>
> - ***(iii)*** Kimi‑K2: Answer: E, Reason: The trailing limb is in late stance with the heel off and only the forefoot pushing, indicating the transition to toe-off that defines pre-swing.
>
> As the above analysis shows, mainstream VLMs can identify gait cycles from silhouettes based on the overall characteristics, which motivates us to leverage textual guidance to learn more universal gait representations.
>
> ---
>
> > [Q3] The predefined gait cycle actions appear biomechanically inspired. What empirical evidence supports that these 8 phases are optimal? Could a data-driven vocabulary discovery approach (e.g., clustering VLM embeddings) yield more discriminative terms? How does performance vary with vocabulary size/granularity?
>
> [A] Thank you for raising these points. The eight phases we adopt correspond to the canonical breakdown of a normal walking cycle established in clinical gait analysis [1, 2] where the statements (“It now is evident that each stride contains 8 functional patterns (phases) [1].” and “In order to describe the elements of gait, the cycle can be broken down further into 8 sub-factors [2].”) demonstrate that eight gait cycle terms potentially capture the spectrum of lower‑limb kinematics and kinetics and is therefore widely accepted as the minimal yet complete description of human walking.
>
> [1] Burnfield M. Gait analysis: normal and pathological function[J]. Journal of Sports Science and Medicine, 2010.
>
> [2] Whittle M W. Gait analysis: an introduction[M]. Butterworth-Heinemann, 2014.
>
> It is worth noting that the eight words are merely a universal starting point, not an upper bound. We provide an alternative way to explore vocabulary size and granularity where we add words that describe common covariates, view angle invariant, bag invariant, and clothing invariant. The results are as follows:
>
> | Model | Vocabulary Base | Rank-1 (%) | mAP (%) |
> |------|------|------|------|
> |Baseline | -   | 74.4   |65.8 |
> | $\alpha$-Gait   | [Initial contact, ..., loading response]   | 76.3   |67.8|
> | $\alpha$-Gait   | [Initial contact, ..., loading response, view-angle invariant, bag invariant, clothing invariant]   | 76.8   |68.4|
>
> As the table shows, we further explore how gait-related terms can potentially affect recognition performance and refine our understanding of gaits. With the flexibility and extensibility of Gait-World, we will continue to explore the vocabularies in future work, thereby deepening understanding within the gait domain.
>
> ---
>
> > [Q4] In line 195, the author points out that "Horizontal Partitioning preserves temporal dimensions", but VRM/GFD processes each frame independently.
>
> [A] Thanks for indicating this question. The statement just highlights that VRM/GFD are applied after Horizontal Partitioning (HP) and process each frame independently. We further clarify this: The gait framework generally contains basic modules: Temporal Pooling (TP) followed by HP. In this work, we only apply HP, which allows the temporal dimension to be preserved before VRM and GFD. We will revise the manuscript to make this clearer.

---

### Official Review · Reviewer_yWT3 · 2025-07-01

**Clarity:** 3
**Significance:** 3
**Originality:** 2
**Rating:** 4
**Confidence:** 5

**Summary:**

This paper introduces vocabulary to the gait recognition, which is a probably new recognition paradigm besides conventional appearance and model-based approaches. Specifically, a cross-attention based architecture, which is referred as α-Gait, is proposed for guiding gait feature learning with vocabulary cues.

**Questions:**

1. α-Gait integrates both components into a unified module. Treating them as separate and simple components diminishes the overall novelty of the method.
2. Many but redundant datasets. Authors can remove some simple datasets such as CASIA-B [1] and add more modality-generalizability experiments.

[1] Shiqi Yu, Daoliang Tan, and Tieniu Tan. A framework for evaluating the effect of view angle, clothing and carrying condition on gait recognition. In 18th International Conference on Pattern Recognition (ICPR’06), volume 4, pages 441–444. IEEE, 2006.

**Ethical Concerns:**

["NO or VERY MINOR ethics concerns only"]

**Final Justification:**

The authors' feedbacks have addressed my concerns on experiments on more challenging datasets. But I feel the novelty is somewhat limited. I therefore keep my initial rating.

**Limitations:**

yes

**Paper Formatting Concerns:**

I haven't noticed major formatting issues.

**Quality:**

2

**Strengths And Weaknesses:**

Strengths:
1. The motivation is clear. Leveraging certain vocabulary bases to explore the potential integration of LLM in gait analysis is worthwhile.
2. The design of vocabulary base is reasonable with respect to the gait cycle.

Weaknesses:
1. The proposed α-Gait lacks of novelty. Although effective, its main components, including Vocabulary Relation Mapper and Gait Fine-grained Detector, are actually existing cross-attention architectures.
2. Vocabulary base lacks of modality-generalizability. The paper repeatedly claims that a-gait can complement existing paradigms such as in lines 237. However, the experiments only present results based on pedestrian silhouettes. The authors are encouraged to validate the generalizability of α-Gait by applying it to other modality like skeleton-maps or point clouds.
3. Computational resource cost. 1) α-Gait introduces additional resource cost. 2) In Table 5, the block numbers of α-Gait are [2, 4, 4, 2] with channels [64, 128, 256, 512], which is larger than most compared methods. Therefore, comparisons of parameters size and FLOPs are recommended.
4. It is better to make code publicly available.

---

> ### Author Rebuttal · Authors · 2025-07-31
>
> Dear Reviewer yWT3:
>
> We greatly value the time and expertise you invested in evaluating our work. Your detailed observations have highlighted important points we can refine, and we address each of them below. Should anything remain unclear, we welcome further discussion.
>
> ---
>
> > [W1] The proposed α-Gait lacks of novelty. Although effective, its main components, including Vocabulary Relation Mapper and Gait Fine-grained Detector, are actually existing cross-attention architectures.
>
> [A] Thanks for your valuable comments. The cross-attention is the partial implementation detail and not the primary contribution. We further clarify the main idea:
>
> - ***(i)*** Gait-World provides a better human understanding of gaits, and a new paradigm to complement existing paradigms (i.e., appearance-based and model-based methods). It serves as an intuitive and efficient tool for researchers to refine their understanding of gait patterns, providing new directions for gait research.
>
> - ***(ii)*** α-Gait designs Vocabulary Relation Mapper (VRM) to align vocabulary features into the gait space and establish associations among vocabularies, addressing feature distribution discrepancy between the Text Encoder and Gait Encoder. Furthermore, α-Gait designs Gait Fine-grained Detector (GFD) to present an object detection perspective where the vocabulary feature detects gait features with the corresponding semantics, which demonstrates the effectiveness of vocabularies. Specifically, the main differences between α-Gait and the attention mechanism are: 1) The objectives. α-Gait aims to provide an intuitive demonstration of the vocabulary’s effectiveness for gait recognition. 2) The implementation. The Query is the text embeddings for guiding the gait network towards universal representations beyond the gait datasets.
>
> In the main manuscript, we will reorganize the contribution to highlight the principal innovation.
>
> ---
>
> > [W2] Vocabulary base lacks of modality-generalizability. The paper repeatedly claims that a-gait can complement existing paradigms such as in lines 237. However, the experiments only present results based on pedestrian silhouettes. The authors are encouraged to validate the generalizability of α-Gait by applying it to other modality like skeleton-maps or point clouds.
>
> [A] Thanks for highlighting the suggestions. We extend α‑Gait from silhouettes to skeleton sequences, and the results confirm that the vocabulary guidance remains effective beyond a single modality. Specifically, we conduct the experiment with SkeletonGait[1] in α-Gait framework on Gait3D：
>
> | Model | Venue | Rank-1 (%) | mAP (%) |
> |------|------|------|------|
> |SkeletonGait | AAAI24  | 37.9   |29.6 |
> | SkeletonGait with α-Gait   | -  | 40.3   |31.3|
> |SkeletonGait++ | AAAI24  | 76.0   |69.2 |
> | SkeletonGait++ with α-Gait   | -  | 78.1   |71.5|
>
> The table shows the reproduction performance where the vocabulary can also be effective in other modalities (i.e., Skeleton maps), which further proves the modality-generalizability.
>
> [1] Fan C, Ma J, Jin D, et al. Skeletongait: Gait recognition using skeleton maps[C]//Proceedings of the AAAI conference on artificial intelligence. 2024.
>
> ---
>
> > [W3] Computational resource cost. 1) α-Gait introduces additional resource cost. 2) In Table 5, the block numbers of α-Gait are [2, 4, 4, 2] with channels [64, 128, 256, 512], which is larger than most compared methods. Therefore, comparisons of parameters size and FLOPs are recommended.
>
> [A] Thanks for this comment. The classification head (e.g., linear layers) scales linearly with the number of identities, overwhelmingly dominating the parameter count in most gait-recognition. Specifically, GREW contains roughly 20k IDs. To highlight differences in the computation cost more clearly, we therefore report comparison results on Gait3D (e.g., smaller IDs about 3k), including α-Gait with [2, 4, 4, 2] and [1, 4, 4, 1] backbones:
>
> | Model | Venue | Rank-1 (%) | mAP (%) | Params. (M) | FLOPs. (G) |
> |------|------|------|------|------|------|
> |GaitBase| CVPR23   | 64.6  |- |19.3 |35.4|
> |GaitGL| ICCV23| -  |- |60.3 |58.6 |
> |DyGait| ICCV23| 66.3  |56.4 |133.1 |239.0|
> |HSTL| ICCV23| 61.3  |55.5 |54.7 |29.7|
> |DeepGaitV2| TPAMI25| 74.4  |65.8 |25.5 |85.3|
> | α-Gait with [2, 4, 4, 2] | -   | -  |- |64.7 |120.9|
> | α-Gait with [1, 4, 4, 1] | -   | 76.3  |67.8 |59.1 |85.6|
>
> The table shows that α-Gait enhances performance without substantially increasing computational cost.
>
> ---
>
> > [W4] It is better to make code publicly available.
>
> [A] Thanks for the suggestion. We will consolidate the implementation and open‑source the complete codebase.
>
> ---
>
> > [Q1] α-Gait integrates both components into a unified module. Treating them as separate and simple components diminishes the overall novelty of the method.
>
> [A] Thank you for the valuable feedback. Similar to the previous weaknesses, we respectfully invite you to see [W1].
>
> ---
>
> > [Q2] Many but redundant datasets. Authors can remove some simple datasets such as CASIA-B and add more modality-generalizability experiments.
>
> [A] Thank you for the constructive suggestion. We will revise our experimental setup and dedicate additional studies to modality generalizability. For example, incorporating skeleton maps experiments as suggested in [W2].

---

> > ### Comment · Reviewer_yWT3 · 2025-08-09
> >
> > Thanks for the authors' feedback, which have addressed some of my concerns on experiments on more challenging datasets. But I feel the novelty is somewhat limited. I therefore keep my initial rating.

---

> > > ### Author Response · Authors · 2025-08-09
> > >
> > > Dear Reviewer yWT3,
> > >
> > > Thank you for your thoughtful follow-up and for acknowledging that our response addresses your concerns.  We appreciate the positive score of your evaluation and the time you invested in our work.
> > >
> > > Regarding novelty, we would like to clarify our core contributions more explicitly:
> > >
> > > (i) Objectives. This work aims to provide an demonstration and extensibility of the vocabulary’s effectiveness for gait recognition.
> > >
> > > (ii) Paradigm. We introduce a vocabulary-driven paradigm Gait-World for gait understanding. It serves as an efficient tool for researchers to refine their understanding of gait patterns, providing new directions for gait research.
> > >
> > > (iii) Method. We present α-Gait with VRM and GFD, acting as the detector that localizes to gait features with the corresponding vocabulary semantics, which provides an intuitive demonstration of the vocabulary’s effectiveness. The cross-attention is the partial implementation detail and not the primary contribution.
> > >
> > > We are grateful for your constructive feedback and will revise the manuscript to highlight these points more clearly. Thank you again for your careful review.
> > >
> > > Sincerely,
> > >
> > > The Authors

---

### Official Review · Reviewer_xBEF · 2025-07-02

**Clarity:** 2
**Significance:** 2
**Originality:** 3
**Rating:** 4
**Confidence:** 4

**Summary:**

The paper introduces a novel paradigm called Gait-World, which aims to improve gait recognition by integrating human-defined vocabulary with Vision-Language Models (VLMs). Existing appearance-based and model-based gait recognition methods suffer from limitations in interpretability and generalization. This work proposes to bridge that gap by leveraging semantic vocabulary related to human gait to guide gait feature learning by VLMs. The proposed method was evaluated on CASIA-B, CCPG, SUSTech1K, Gait3D and GREW.

**Questions:**

(1) Gait recognition requires fine-grained details, why the general phases text can help the gait recognition?
(2) Gait of different people have same routine (initial contact, loading responses, ...), why the text can lead to better gait recognition performance?
(3) Is it feasible to extend the gait vocabulary to further boost the performance?
(4) Can the model take text related to appearance such as clothing, carrying, etc.

**Ethical Concerns:**

["NO or VERY MINOR ethics concerns only"]

**Final Justification:**

The rebuttal addressed most of my concerns. I keep my original rating.

**Limitations:**

yes

**Paper Formatting Concerns:**

No major formatting issues

**Quality:**

2

**Strengths And Weaknesses:**

Strengths:
(1) This paper introduces a new framework that uses human-understandable vocabulary (gait cycle phases) to guide gait representation learning.
(2) By leveraging a text encoder, the semantic information may help the gait encoder learn better features.
(3) Alpha-gait achieves strong results on benchmark datasets
(4) The paper includes adequate ablation studies of the effect of vocabulary guidance, VLM type, and network components.

Weaknesses:
(1) The predefined vocabulary base only contain eight terms, which may not capture various gait patterns.
(2) Despite framing the work as vocabulary-guided, all inputs share the same fixed set of textual prompts.
(3) The encoder such as CLIP was not pretrained on gait-related data such as eilhouette, which may not align well with the words in the vocabulary.
(4) There is no verification of semantic alignment of gait features and texts

---

> ### Author Rebuttal · Authors · 2025-07-31
>
> Dear Reviewer xBEF:
>
> Thank you for your insightful feedback and thoughtful suggestions. Below we respond point‑by‑point, clarifying our methods and results where needed. Please feel free to let us know if any issues remain.
>
> ---
>
> > [W1] The predefined vocabulary base only contains eight terms, which may not capture various gait patterns.
>
> [A] Thank you for the thoughtful comment. We address this concern from three perspectives:
>
> - ***(i)*** The reason for vocabulary selection. The statements in clinical gait literatures [1, 2] (“It now is evident that each stride contains 8 functional patterns (phases) [1].” and “In order to describe the elements of gait, the cycle can be broken down further into 8 sub-factors [2].”) consistently partitions a normal stride into eight distinct phases. Hence, using eight gait‑cycle terms is widely accepted as the minimal yet comprehensive way to describe lower‑limb kinematics and kinetics during walking.
>
>     [1] Burnfield M. Gait analysis: normal and pathological function[J]. Journal of Sports Science and Medicine, 2010.
>
>     [2] Whittle M W. Gait analysis: an introduction[M]. Butterworth-Heinemann, 2014.
>
> - ***(ii)*** More kinds of words. Since gait recognition needs to stay robust to common covariates (e.g., cross‑view, cross‑bag and cross‑clothing conditions), we additionally experimented with more vocabularies: “view-angle invariant”, “bag invariant”, “clothing invariant”, and we analyze the effect as reported:
>
>     | Model | Vocabulary Base | Rank-1 (%) | mAP (%) |
>     |------|------|------|------|
>     |Baseline | -   | 74.4   |65.8 |
>     | $\alpha$-Gait   | [Initial contact, ..., loading response]   | 76.3   |67.8|
>     | $\alpha$-Gait   | [Initial contact, ..., loading response, view-angle invariant, bag invariant, clothing invariant]   | 76.8   |68.4|
>
>     As the table shows, we further explore how gait-related terms can potentially affect recognition performance and refine our understanding of gaits. With the flexibility and extensibility of Gait-World, we will continue to explore the vocabularies in future work, thereby deepening understanding within the gait domain.
>
> - ***(iii)*** The extraction of gait patterns. This work treats these words as semantic queries to compare different gait sequences. Therefore, the model will learn fine‑grained gait patterns adaptively from the gait data, preserving the diversity of gait patterns.
>
> ---
>
> > [W2] Despite framing the work as vocabulary-guided, all inputs share the same fixed set of textual prompts.
>
> [A] Thank you for highlighting this issue. We intentionally keep the vocabulary fixed for all samples:
>
> - ***(i)*** Unified semantics for fine-grained representations. A shared vocabulary base anchors every sequence to the same semantic directions, allowing the visual encoder to capture the precise and fine‑grained identity cues.
>
> - ***(ii)*** Text Guidance against appearance noises. Gait recognition needs to capture identity from the complex environment, such as view angles and clothes. A consistent textural prompt acts as an external prior, guiding the network toward identity‑specific information rather than appearance‑based confounders.
>
> ---
>
> > [W3] The encoder such as CLIP was not pretrained on gait-related data such as silhouette, which may not align well with the words in the vocabulary.
>
> [A] Thank you for raising this issue. We address the concern from two perspectives:
>
> - ***(i)*** The modal alignment is learnable. The Vocabulary Relation Mapper (VRM) and Gait Fine-grained Detector (GFD) are jointly optimized under an identity‑supervised loss, which forces the silhouette features to align with the eight textual directions, so recognition does not only depend on the text encoder.
>
> - ***(ii)*** The text space within VLMs exhibits strong universal capability. Although CLIP’s pre‑training set lacks silhouettes and Llama is a text‑only framework, their text embeddings still can construct general human‑motion concepts (e.g., “initial swing,” “loading response”). During training, the visual encoder learns the silhouette patterns with these universal axes, giving the network covariate‑invariant features that extend beyond the gait training set.
>
> ---
>
> > [W4] There is no verification of semantic alignment of gait features and texts?
>
> [A] Thanks for your valuable comments. We visualize the process of vocabulary guidance in Figure 4 of the main manuscript, where GFD detects the eight silhouettes with the corresponding eight gait cycle vocabularies. Additionally, we further conduct more visualizations that consistently corroborate the alignment between gait features and their textual counterparts. Owing to the latest NeurIPS rebuttal constraints (No permission for the additional images or links), we will present more results in the revised manuscript.
>
> ---
>
> > [Q1] Gait recognition requires fine-grained details, why the general phases text can help the gait recognition?
>
> [A] Thanks for this valuable question. Similar to the previous weaknesses, we respectfully invite you to see [W1, W2].
>
> ---
>
> > [Q2] Gait of different people have same routine (initial contact, loading responses, ...), why the text can lead to better gait recognition performance?
>
> [A] Thanks for raising this important point. Similar to the previous weaknesses, we respectfully invite you to see [W1, W2].
>
> ---
>
> > [Q3] Is it feasible to extend the gait vocabulary to further boost the performance?
>
> [A] Thanks for this suggestion. We explore vocabulary expansion on top of the eight gait cycle terms and observe consistent improvements. Similar to the previous weakness, we respectfully invite you to see [W1].
>
> ---
>
> > [Q4] Can the model take text related to appearance such as clothing, carrying, etc.
>
> [A] Thanks for the valuable comment. We add appearance vocabularies: “view-angle invariant”, “bag invariant”, “clothing invariant”. Similar to the previous weakness, we respectfully invite you to see [W1].

---

> > ### Comment · Reviewer_xBEF · 2025-08-05
> > **Concerns addrssed**
> >
> > Thanks authors for the rebuttal. Most of my concerns have been addressed. I lean to accepting the paper.

---

> ### Author Response · Authors · 2025-08-05
>
> Dear Reviewer xBEF,
>
> Thank you for your thoughtful follow-up. We are delighted that the response has addressed your main concerns, and we greatly appreciate your positive inclination toward acceptance. Your constructive feedback has been helpful in improving our work.
>
> Best regards,
>
> The Authors

---

### Official Review · Reviewer_VrEo · 2025-07-03

**Clarity:** 4
**Significance:** 3
**Originality:** 4
**Rating:** 5
**Confidence:** 5

**Summary:**

This paper has a new idea. It uses words about walking, like 'initial contact' or 'mid swing', to help computers recognise people from their walk (gait). They call this new method Gait-World. Their model, α-Gait, takes these words from a Vision-Language Model (VLM) and uses them to guide the main gait recognition network. This helps the network focus on the right parts of the walking style. They tested it on many datasets like CASIA-B, CCPG, and GREW, and showed that their model works very well, even better than many old methods.

**Questions:**

1. Your Vocabulary Base has only eight phrases, which are the technical terms for a gait cycle.  Why did you choose only these? Did you try using more simple or descriptive words like "walking fast", "limping", or "carrying a bag"?
2. α-Gait model uses a Text Encoder which is a big model.  How much does this slow down the recognition time compared to a normal model like GaitBase? Can you give some numbers on inference speed?
3. Your results on the CCPG dataset are much better than other methods , but on CASIA-B the improvement is not so big.  Can you explain why your method works so well for clothes-changing but is only competitive in normal view-changing scenarios?
4. Gait recognition is a biometric technology and can be used for surveillance. This has a negative societal impact. Can you please add a discussion about these risks and how they can be managed?

**Ethical Concerns:**

["NO or VERY MINOR ethics concerns only"]

**Final Justification:**

After author feedback, i would keep my rating of accept.

**Limitations:**

No. The authors have a small section on limitations , but it is only about future technical work, like using more words or making it a multimodal system.  They do not at all discuss the main limitation which is the potential negative societal impact of gait recognition technology (e.g., surveillance). This is a serious miss.

**Paper Formatting Concerns:**

The paper formatting is fine. No issues.

**Quality:**

3

**Strengths And Weaknesses:**

Strengths:

Originality is very high. The idea to use human vocabulary to guide gait recognition is very new. I have not seen this before. It connects two different fields, language and vision, for this problem.

Significance is good. This paper opens a new way to do research in gait recognition. Other people can now try using different words or better language models to improve results.

Quality of experiments is strong. The authors tested their model α-Gait on five big datasets (CASIA-B, CCPG, SUSTech1K, Gait3D, GREW). The results in the tables show it is better than many state-of-the-art methods, especially in difficult situations like changing clothes (CCPG dataset). The ablation studies are also good to show why their idea works.

Clarity is good. The paper is written well. It is easy to understand the main idea. Figure 3 and Figure 4 are very helpful to see how the model works and what it is learning.

Weaknesses:

The

Vocabulary Base is very small, only eight words for the gait cycle.  The paper does not say if they tried more words or different kinds of words. This feels a bit limited.

The paper does not talk about the bad side of this technology. Gait recognition can be used for surveillance and tracking people without them knowing. This is a big ethical problem and the authors should have discussed it. In the checklist, they just said Not Applicable for societal impact, which is not right.
The model architecture for α-Gait is quite simple. It uses standard blocks like attention. The main new thing is the idea (Gait-World), not a new complex model design.

---

> ### Author Rebuttal · Authors · 2025-07-31
>
> Dear Reviewer VrEo:
>
> We greatly appreciate your careful assessment and constructive insights. We provide detailed point‑by‑point responses below. Please let us know if any additional clarification is needed.
>
> ---
>
> > [W1] The Vocabulary Base is very small, only eight words for the gait cycle. The paper does not say if they tried more words or different kinds of words. This feels a bit limited.
>
> [A] Thanks for your valuable comments. We further explain this as follows:
>
> - ***(i)*** The reason for vocabulary selection. The eight phases we adopt correspond to the canonical breakdown of a normal walking cycle established in clinical gait analysis [1, 2] where the statements (“It now is evident that each stride contains 8 functional patterns (phases) [1].” and “In order to describe the elements of gait, the cycle can be broken down further into 8 sub-factors [2].”) demonstrate that eight gait cycle terms potentially capture the spectrum of lower‑limb kinematics and kinetics and is therefore widely accepted as the minimal yet complete description of human walking.
>
>     [1] Burnfield M. Gait analysis: normal and pathological function[J]. Journal of Sports Science and Medicine, 2010.
>
>     [2] Whittle M W. Gait analysis: an introduction[M]. Butterworth-Heinemann, 2014.
>
> - ***(ii)*** More kinds of words. The gait network needs to extract representations that are invariant to complex covariates (e.g., cross-view, cross-bag and cross-clothing conditions). Therefore, we try to explore the vocabularies that describe common covariates: “view-angle invariant”, “bag invariant”, “clothing invariant” and evaluate the impact as follows:
>
>     | Model | Vocabulary Base | Rank-1 (%) | mAP (%) |
>     |------|------|------|------|
>     |Baseline | -   | 74.4   |65.8 |
>     | $\alpha$-Gait   | [Initial contact, ..., loading response]   | 76.3   |67.8|
>     | $\alpha$-Gait   | [Initial contact, ..., loading response, view-angle invariant, bag invariant, clothing invariant]   | 76.8   |68.4|
>
>     As the table shows, we further explore how gait-related terms can potentially affect recognition performance and refine our understanding of gaits. With the flexibility and extensibility of Gait-World, we will continue to explore the vocabularies in future work, thereby deepening understanding within the gait domain.
>
> - ***(iii)*** The extensibility for Gait-World. We use universal eight gait cycle terms to highlight the effectiveness of vocabularies. However, Gait-World serves as an intuitive and efficient way for researchers to refine their understanding of gait patterns, not limited to these words.
>
> ---
>
> > [W2] The paper does not talk about the bad side of this technology. Gait recognition can be used for surveillance and tracking people without them knowing. This is a big ethical problem and the authors should have discussed it. In the checklist, they just said Not Applicable for societal impact, which is not right.
>
> [A] Thank you for raising this crucial issue. Safeguarding privacy remains our top priority as we advance AI research. This work discusses the ethical problem from three perspectives:
>
> - ***(i)*** This work exclusively studies publicly available datasets whose participants granted prior consent for academic use, and only silhouette data are used, suppressing conspicuous visual cues (e.g., faces).
>
> - ***(ii)*** We agree that gait recognition, like many biometric technologies, carries a real risk of misuse for covert surveillance and mass tracking. We firmly believe that gait recognition technology should be deployed only with explicit, informed consent or under a clearly defined legal mandate, and never for indiscriminate, covert mass surveillance.
>
> - ***(iii)*** We will add a dedicated ethics section and update the checklist accordingly.
>
> ---
>
> > [W3] The model architecture for α-Gait is quite simple. It uses standard blocks like attention. The main new thing is the idea (Gait-World), not a new complex model design.
>
> [A] Thank you for the thoughtful analysis. We reorganize and highlight the main idea as follows:
>
> - ***(i)*** Gait-World provides a better human understanding of gaits and a new paradigm to complement existing paradigms. It serves as an intuitive and efficient tool for researchers to refine their understanding of gait patterns, providing new directions for gait research.
>
> - ***(ii)*** α-Gait designs Vocabulary Relation Mapper (VRM) to align vocabulary features into the gait space and establish associations among vocabularies, addressing feature distribution discrepancy between the Text Encoder and Gait Encoder. Furthermore, α-Gait designs the Gait Fine-grained Detector (GFD) to present an object detection perspective where the vocabulary feature detects gait features with the corresponding semantics, which demonstrates the effectiveness of vocabularies. Specifically, the main differences between α-Gait and the attention mechanism are: 1) The objectives. α-Gait aims to provide an intuitive demonstration of the vocabulary’s effectiveness for gait recognition. 2) The implementation. The Query is the text embeddings for guiding the gait network towards universal representations beyond the gait datasets.
>
> In the main manuscript, we will reorganize the contribution to highlight the principal innovation.
>
> ---
>
> > [Q1] Your Vocabulary Base has only eight phrases, which are the technical terms for a gait cycle. Why did you choose only these? Did you try using more simple or descriptive words like "walking fast", "limping", or "carrying a bag"?
>
> [A] Thank you for the thoughtful comments. Similar to the previous weakness, we respectfully invite you to see [W1].
>
> ---
>
> > [Q2] α-Gait model uses a Text Encoder that is a big model. How much does this slow down the recognition time compared to a normal model like GaitBase? Can you give some numbers on inference speed?
>
> [A] Thank you for raising this point. Since the vocabulary is fixed and the Text Encoder is frozen, α-Gait can pre‑compute and cache the word embeddings to largely reduce the computation. The relative statistics are as follows:
>
> | Model | Venue | Rank-1 (%) | mAP (%) | Params. (M) | FLOPs. (G) |
> |------|------|------|------|------|------|
> |GaitBase| CVPR23   | 64.6  |- |19.3 |35.4|
> |GaitGL| ICCV23| -  |- |60.3 |58.6 |
> |DyGait| ICCV23| 66.3  |56.4 |133.1 |239.0|
> |HSTL| ICCV23| 61.3  |55.5 |54.7 |29.7|
> |DeepGaitV2| TPAMI25| 74.4  |65.8 |25.5 |85.3|
> | $\alpha$-Gait| -   | 76.3  |67.8 |59.1 |85.6|
>
> The table shows that α-Gait enhances performance without substantially increasing computational cost.
>
> ---
>
> > [Q3] Your results on the CCPG dataset are much better than other methods, but on CASIA-B, the improvement is not so big. Can you explain why your method works so well for clothes-changing but is only competitive in normal view-changing scenarios?
>
> [A] Thank you for highlighting this point.  We observe two complementary causes:
>
> - ***(i)*** Appearance‑invariant vocabulary. The eight gait‑cycle words describe walking phases (e.g., heel‑strike, toe‑off, etc.), which are intrinsically independent of the appearance. They therefore retain identity cues when apparel changes drastically, giving a larger relative gain on clothes‑changing benchmarks.
>
> - ***(ii)*** Saturated CASIA-B recognition performance. CASIA‑B changes only upper‑body clothes. Therefore, the legs remain visually consistent in CASIA‑B and appearance‑based methods already extract stable motion cues, saturating the recognition performance (e.g., 98% on Normal Walking). In contrast, CCPG includes full‑body, partial and backpack variations, which severely confound appearance‑based methods, while our walk‑centric embeddings stay robust and thus yield a conspicuous margin.
>
> ---
>
> > [Q4] Gait recognition is a biometric technology and can be used for surveillance. This has a negative societal impact. Can you please add a discussion about these risks and how they can be managed?
>
> [A] Thank you for highlighting this important issue. In addition to the covert mass surveillance and informed consent concerns noted in [W2], we further discuss that gait recognition, as a remote biometric, could be misused for algorithmic bias and function creep. Below, we outline the primary risks and the concrete safeguards:
>
> - ***(i)*** Risk 1: Covert mass surveillance
>
>     Technical safeguards: Release only binary silhouettes / skeletons (no RGB frames) and an explicit ban on live‑CCTV deployment without informed consent.
>
> - ***(ii)*** Risk 2: Informed consent
>
>     Technical safeguards: The data are released strictly for non‑commercial research, subject to case‑by‑case approval. If an algorithm nevertheless recovers a subject’s identity, that identity is deleted immediately and permanently. License requires erasure of recovered identities and mandatory reporting within 48 hours of any breach.
>
> - ***(iii)*** Risk 3: Algorithmic bias
>
>     Technical safeguards: Provide sub‑split test metrics (gender, age, assistive devices); if inequality‑deviation rate exceeds a threshold, retrain with re‑weighting and fairness regularization. Yearly bias reports in the model card for community review.
>
> - ***(iv)*** Risk 4: Function creep
>
>     Technical safeguards: Data and models released only after ethics board approval. Access will be immediately revoked, and any violations will be publicly disclosed if the technology is deployed beyond the approved scope.
>
> ---
>
> > [L1] The authors have a small section on limitations, but it is only about future technical work, like using more words or making it a multimodal system. They do not at all discuss the main limitation which is the potential negative societal impact of gait recognition technology (e.g., surveillance). This is a serious miss.
>
> [A] Thank you for pointing out this gap. Similar to the previous weakness and question, we respectfully invite you to see [W2] and [Q4]. We will expand the Limitations & Societal Impact to address the ethical risks of gait recognition.

---

> > ### Comment · Reviewer_VrEo · 2025-08-05
> > **Satisfied with the rebuttal**
> >
> > Thanks for the rebuttal. Most of my concerns have been addressed.

---

> > > ### Author Response · Authors · 2025-08-06
> > >
> > > Dear Reviewer VrEo,
> > >
> > > We sincerely appreciate the time and effort you dedicated to reviewing our paper. We’re very pleased that your main concerns are satisfied with our response. Your insightful suggestions have substantially improved our work.
> > >
> > > Best regards,
> > >
> > > The Authors

---

### Note · Authors · 2025-08-12

Dear ACs and Reviewers,

Thank you for your time, thoughtful feedback, and constructive suggestions to review and improve our work, and we greatly appreciate your positive inclination toward acceptance. **All reviewers maintained positive ratings from the initial stage through the post-rebuttal discussion.**

Our rebuttal and experiments mainly resolve the following concerns:

**(i) Extensibility and Soundness of the Vocabulary Base.** We provide substantial references supporting the rationale for using the eight canonical gait phases as a minimal, complete vocabularies. Extending it with view-angle, bag, and clothing invariants further improves performance, confirming extensibility.

**(ii)	Clarification of Novelty.** (a) Gait-World provides a better human understanding of gaits, and a new paradigm to complement existing paradigms (i.e., appearance-based and model-based methods). It serves as an intuitive and efficient tool for researchers to refine their understanding of gait patterns, providing new directions for gait research. (b) α-Gait provides a demonstration of the vocabulary’s effectiveness for gait recognition via the Vocabulary Relation Mapper (VRM) and Gait Fine-grained Detector (GFD), which align and relate vocabulary and gait.

**(iii) Modality- and Architecture-agnostic Validation.** We extend α‑Gait from silhouettes to skeleton sequences, confirming effectiveness beyond a single modality (SkeletonGait vs. SkeletonGait with α‑Gait: Rank-1 accuracy from 37.9% to 40.3% on Gait3D), and pair it with SkeletonGait++ to demonstrate robustness across architectures (SkeletonGait++ vs. SkeletonGait++ with α‑Gait: Rank-1 accuracy from 76.0% to 78.1% on Gait3D).

**(iv) Efficiency.** α-Gait freezes the text encoder and caches vocabulary embeddings, preserving efficiency. Parameters/FLOPs comparisons show accuracy gains without substantial overhead.

**(v) Responsible Ethics.** We discuss key risks: covert mass surveillance, informed consent, algorithmic bias, and function creep, and outline concrete safeguards and privacy protections.

Gait-World aims to provide a better human understanding of gaits, and a new paradigm to complement existing paradigms. α-Gait provides an intuitive demonstration of the vocabulary’s effectiveness for gait recognition.

We are sincerely grateful for your efforts and the constructive discussion. Thank you once again for your support.

Best regards,

The Authors

---

### Decision · Program_Chairs · 2025-09-17

**Decision:**

Accept (poster)

**Comment:**

This paper introduces Gait-World and the α-Gait model, a novel paradigm that leverages human-interpretable vocabulary with vision-language models to guide gait recognition. The motivation is clear, and the approach offers a fresh perspective that bridges language and vision for biometric analysis. Experiments across five benchmark datasets demonstrate strong performance, particularly under challenging conditions such as clothing changes, with solid ablations validating the design choices.

Reviewers appreciated the originality, clarity, and empirical strength of the work, though some raised concerns about the limited vocabulary base, potential societal impacts of gait recognition, and the relatively simple model design. The rebuttal addressed these points by providing further justification of the vocabulary selection, evidence of extensibility to other words and modalities, efficiency analysis, and an explicit discussion of ethical risks and safeguards. Most reviewers were satisfied with the responses and several leaned more positively after rebuttal.

Overall, the paper presents a technically solid and original contribution with significant potential impact. I recommend acceptance.